# Systematic identification of functional SNPs interrupting 3'UTR polyadenylation signals

**Eldad David Shulman**, **Ran Elkon** *

Department of Human Molecular Genetics and Biochemistry, Sackler School of Medicine, Tel Aviv University, Tel Aviv, Israel

* ranel@tauex.tau.ac.il

**Data Availability Statement:** All relevant data are within the manuscript and its Supporting Information files.

**Funding:** R.E. Israel Science Foundation grant no. 2118/19 DIP German–Israeli Project cooperation (DFG RE 4193/1-1), Bernard Jacobson's

## Abstract

Alternative polyadenylation (APA) is emerging as a widespread regulatory layer since the majority of human protein-coding genes contain several polyadenylation (p(A)) sites in their 3'UTRs. By generating isoforms with different 3'UTR length, APA potentially affects mRNA stability, translation efficiency, nuclear export, and cellular localization. Polyadenylation sites are regulated by adjacent RNA cis-regulatory elements, the principals among them are the polyadenylation signal (PAS) AAUAAA and its main variant AUUAAA, typically located ~20-nt upstream of the p(A) site. Mutations in PAS and other auxiliary poly(A) cis-elements in the 3'UTR of several genes have been shown to cause human Mendelian diseases, and to date, only a few common SNPs that regulate APA were associated with complex diseases. Here, we systematically searched for SNPs that affect gene expression and human traits by modulation of 3'UTR APA. First, focusing on the variants most likely to exert the strongest effect, we identified 2,305 SNPs that interrupt the canonical PAS or its main variant. Implementing pA-QTL tests using GTEx RNA-seq data, we identified 330 PAS SNPs (called PAS pA-QTLs) that were significantly associated with the usage of their p(A) site. As expected, PAS-interrupting alleles were mostly linked with decreased cleavage at their p(A) site and the consequential 3'UTR lengthening. However, interestingly, in ~10% of the cases, the PAS-interrupting allele was associated with increased usage of an upstream p(A) site and 3'UTR shortening. As an indication of the functional effects of these PAS pA-QTLs on gene expression and complex human traits, we observed for few dozens of them marked colocalization with eQTL and/or GWAS signals. The PAS-interrupting alleles linked with 3'UTR lengthening were also strongly associated with decreased gene expression, indicating that shorter isoforms generated by APA are generally more stable than longer ones. Last, we carried out an extended, genome-wide analysis of 3'UTR variants and detected thousands of additional pA-QTLs having weaker effects compared to the PAS pA-QTLs.

## Author summary

mRNA molecules that encode for proteins end with a long stretch of adenosines, called poly(A) tail. The poly(A) tail contributes to the stability of the mRNA molecules, their translation to proteins and their import from the nucleus to the cytoplasm. The process of

fund - TAU, ED.S Edmond J. Safra Center for Bioinformatics at Tel Aviv University. The funders had no role in study design, data collection and analysis, decision to publish, or preparation of the manuscript.

**Competing interests:** The authors have declared that no competing interests exist.

adding this tail to the mRNAs is called polyadenylation, and the termination site on the mRNAs at which the poly(A) tail is added is called the poly(A) site. In recent years it became evident that the vast majority of mRNAs of human genes contain several alternative poly(A) sites and their usage generates different mRNA isoforms that differ in their stability and translation efficiency. Therefore, alternative polyadenylation (APA) is emerging as a novel and important, yet underexplored, mechanism that regulate gene expression. The choice between alternative p(A) sites in an mRNA molecule is regulated by regulatory sequences located within a region in the mRNA called the 3' untranslated region (3'UTR). A major challenge in present human genetics research is to understand how common genetic variants affect individuals' health. In our study, we systematically identified dozens of genetic variants that affect the choice between alternative p(A) sites and demonstrated that by that, these variants influence the expression level of the target genes. Our results help to illuminate a novel mechanism by which genetic variants that are common in the population affect different traits including our risk for developing diseases.

## Introduction

The maturation of mRNA 3′ ends is a 2-steps process, termed *cleavage and polyadenylation*, that involves endonucleolytic cleavage of the nascent RNA followed by synthesis of a poly(A) tail on the 3′ terminus of the cleaved product [1]. Cleavage and polyadenylation sites (p(A) sites) are determined and controlled by adjacent RNA cis-regulatory elements, the principal among them is the *polyadenylation signal* (*PAS*) AAUAAA, typically located ~20-nt upstream of the p(A) site. There are more than 10 weaker variants to this canonical PAS, the main among them is AUUAAA [2]. Auxiliary elements include upstream U-rich and UGUA motifs and downstream U-rich and GU-rich elements, and the strength of a p(A) site is determined by these elements in a combinatorial manner [3]. Importantly, the majority of human protein-coding genes contain several alternative p(A) sites in their 3'UTR, making alternative polyadenylation (APA) a widespread regulatory layer that generates transcript isoforms with alternative 3′ ends, and correspondingly, different 3'UTR lengths [1, 4–6]. As 3′ UTRs contain cis-elements that serve as major docking platforms for microRNAs (miRNAs) and RNA binding proteins (RBPs), which are involved in various aspects of mRNA metabolism, 3′ UTR APA can affect post-transcriptional regulation in multiple ways, including the modulation of mRNA stability, translation efficiency, nuclear export and cellular localization [7–9]. Transcriptomic studies demonstrated that APA is globally modulated during multiple differentiation processes [4, 10–17] and in response to changes in cell proliferation state [18–20]. Yet, our current understanding of the impact of APA on gene regulation and of its biological roles is still very rudimentary.

Mutations in PAS and other poly(A) cis-elements, and the resulting alteration in gene expression have been shown to cause several human Mendelian diseases. Examples include the mutation in the 3'UTR of *HBA2* (converting AATAAA to AATAAG) causing α-Thalassaemia [21], the mutation in the 3'UTR of *HBB* (AATAAA to AACAAA) causing β-Thalassaemia [22] and the mutation in the 3'UTR of *FOXP3* (AATAAA to AATGAA) causing the IPEX syndrome [23] (for a thorough review see [24]). In addition, few common SNPs that regulate APA were found to affect the risk of complex diseases. This includes a risk SNP for systemic lupus erythematosus (SLE) located in the 3'UTR of *IRF5*. This SNP reduces the use of a proximal p(A) site, leading to the production of longer and less stable isoforms and consequently to

reduced *IRF5* levels [25]. Similarly, a polymorphic cis-element downstream to a PAS in the 3'UTR of *ATP1B1* is associated with high blood pressure [26].

Given the high prevalence of APA, there are likely many more common SNPs that modulate APA and affect disease susceptibility. In this study, we systematically searched for SNPs that interfere with 3'UTR PAS signals in the human genome and implemented polyadenylation-QTL (pA-QTL) tests to examine their effect on APA. Analyzing GTEx RNA-seq data from twelve tissues we identified dozens of such SNPs that significantly affect the usage of their downstream p(A) sites. The intersection of these pA-QTLs with eQTLs and GWAS risk SNPs indicated their roles in the regulation of gene expression and their impact on various phenotypes.

## Results

### pA-QTL analysis of PAS SNPs

We first sought to systematically identify SNPs in the human genome that affect 3'UTR APA. To focus our analysis on the ones likely to exert the strongest effect on gene regulation (and thus on phenotype), we specifically searched for SNPs that interrupt the canonical PAS (AATAAA) or its main variant (ATTAAA). Using p(A) sites annotations from polyA DB [27], we overall detected 2,305 SNPs that interfere with these signals in a 40-nt window upstream of annotated 3'UTR p(A) sites in 1,936 unique genes (Fig 1A; S1 Table). We call such variants *PAS SNPs*. Each PAS SNP has one allele that preserves the PAS and another allele that interrupts it (called PAS Interrupting Allele (*PIA*)). Note that the PIA can be either the allele appearing on the genome reference sequence (the reference allele) or the alternative allele (Fig 1B). 68% of the PAS SNPs were located within the canonical AATAAA PAS (S1 Table). We associated each PAS SNP with its downstream p(A) site.

PAS SNPs potentially have a strong impact on cleavage and polyadenylation efficiency at their downstream p(A) sites. To detect such effects we implemented pA-QTL tests that are based on the estimation of p(A)-site usage from RNA-seq data. Each PAS SNP divides its 3'UTR into two segments: a common UTR (cUTR) which is upstream to the associated p(A) site and is common to the transcript generated by cleavage at this p(A) site and the transcripts generated by usage of more distal p(A) sites, and an alternative UTR (aUTR) which is included only in transcripts generated by more distal p(A) sites (Fig 2A). For each RNA-seq sample, we estimated the usage of each p(A) site associated with a PAS SNP by calculating the p(A) site Usage Index (*pAUI*), defined as the ratio (in log scale) between the number of reads mapped to the cUTR and the aUTR segments (Fig 2A). Having RNA-seq data from a large cohort of individuals for which genotype data is available too, enables to test for association between the PAS SNP alleles and the pAUI levels (Fig 2A). We carried out such pA-QTL tests on twelve selected tissues from the GTEx project (v7) [28] that have more than 130 RNA-seq samples with corresponding genotype data (Methods; S2 Table). Per tissue, and for each p(A) site linked with a PAS SNP, we examined variants within a window of +/- 10 kbp with respect to the p(A) site for association with pAUI levels using FastQTL [29] (Methods). Overall, 512 pA sites were significantly associated (FDR ≤ 5%) with a pA-QTL in at least one tissue (see examples in Fig 2B–2D; S3 Table).

Linkage disequilibrium (LD) patterns between adjacent genetic variants complicate the identification of causal variants. Therefore, next, we used CAVIAR [30] to detect pA-QTL associations that can be more confidently ascribed to the effect of the corresponding PAS SNPs (S1 Fig; Methods). Defining per pA site with a significant association a 95% credible set of variants, this analysis implicated the PAS SNP as a causal variant for 55%-70% of the observed pA-QTLs per tissue (Fig 3A). Hereafter, we refer to PAS SNPs as PAS pA-QTLs if

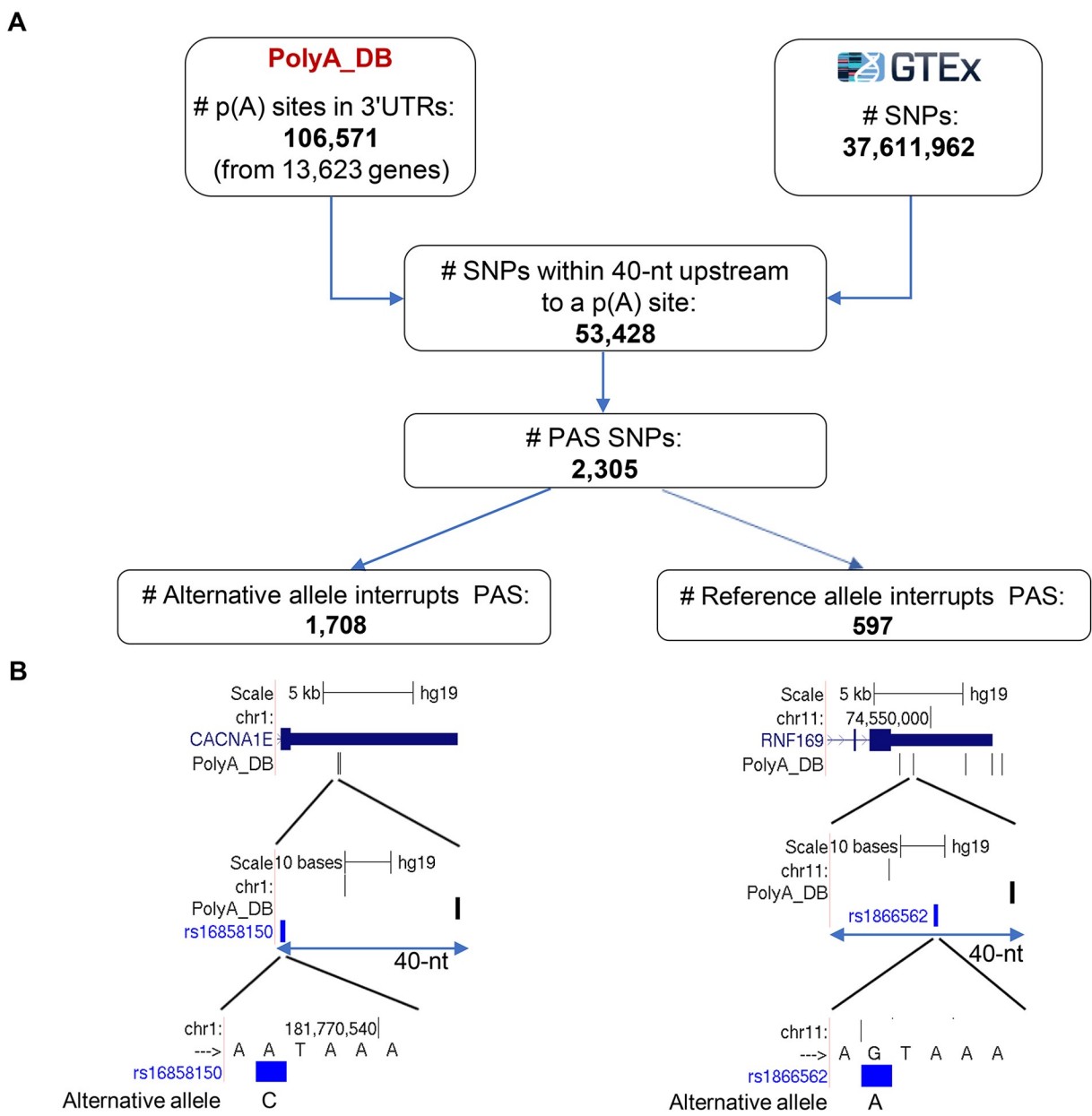

**Fig 1. Systematic identification of PAS SNPs in the human genome. A**. We defined as *PAS SNPs* those that are located within 40-nt upstream of an annotated 3'UTR p(A) site and have an allele that interrupts the canonical PAS sequence AATAAA or its main variant ATTAAA. We considered all the 3'UTR p(A) sites annotated in poly(A) DB (release 3.2), and all ~37M SNPs included in GTEX v7. We detected 2,305 such SNPs. Each biallelic SNP has a reference allele (the allele that appears in the genome's reference sequence) and an alternative allele. Among the 2,305 PAS SNPs detected by our screen, 1,708 SNPs have the alternative allele interrupting the PAS sequence and 597 SNPs have the reference allele disrupting the signal. **B**. An example of a PAS SNP whose alternative allele interrupts the PAS signal (rs16858150 in the 3'UTR of *CACNA1E*; left) and a PAS SNP whose reference allele interrupts it (rs1866562 in the 3'UTR of *RNF169*; right).

they (a) obtained in the pA-QTL tests p-values below their pA site-level thresholds (see Methods) and (b) were included in CAVIAR's credible sets. Overall, we detected 330 distinct PAS pA-QTLs, 70% of them interrupt a canonical PAS (and the rest interrupt the ATTAAA variant) (S3 Table). Most of these PAS pA-QTLs were detected in multiple tissues (Fig 3B and S3 Table). The p(A) loci in which the PAS SNP was not included in CAVIAR's credible set

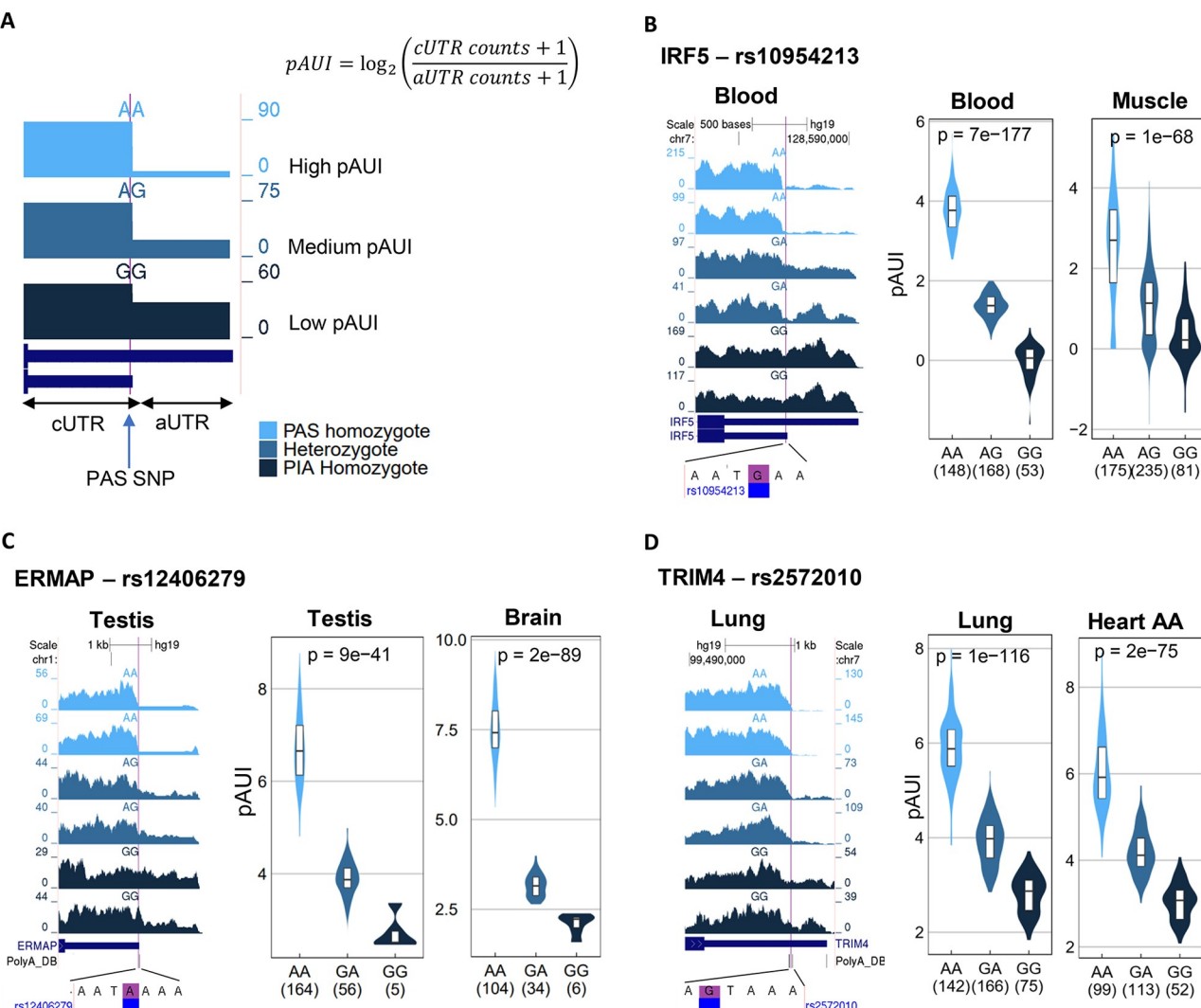

**Fig 2. pA-QTL analysis. A**. We used the p(A) site usage index (pAUI) to quantify cleavage efficiency at each annotated 3'UTR pA site in each RNA-seq sample. The pAUI is defined as the ratio (in log2 scale) between the counts of 3'UTR reads mapped upstream of the pA site (common 3'UTR segment; cUTR) and those mapped downstream of it (alternative 3'UTR segment; aUTR) (Methods). We then used this index to detect PAS SNPs that show a significant association between alleles and pAUI levels of their p(A) site. SNPs showing such association are referred to as pA-QTLs. The expected pattern, as shown in this cartoon, is that the PAS-preserving allele is associated with higher usage of the p(A) site while the PAS-interrupting allele (*PIA*) is associated with reduced usage of this site (resulting in 3'UTR lengthening). Heterozygotes for such SNPs are expected to show intermediate pAUI levels compared to the two homozygotes. The cartoon illustrates reads coverage on a 3'UTR for three RNA-seq samples of varying levels of pAUI, colored according to the genotype of the PAS SNP. **B–D.** Examples of three PAS SNPs consistently detected as pA-QTLs in multiple tissues. In each example, the left panel shows read coverage in the gene's 3'UTR from RNA-seq samples of two selected donors from each PAS SNP genotype. The vertical purple line marks the location of the PAS SNP. The genome reference sequence around the PAS SNP is shown below. Violin plots in the middle and left panels show the distribution of pAUI levels in each genotype group for a given tissue (In each plot, homozygotes of the PAS-preserving allele are shown in the left, heterozygotes–in the middle, and homozygotes to the PAS-interrupting allele (PIA)–in the right violin. The number of individuals in each group is indicated in parentheses, shown are nominal p-values obtained using FastQTL linear regression as described in the Methods). (In C, "Brain" refers to "Brain caudal nucleus", and in D, "Heart AA" refers to "Heart atrial appendage").

indicate that there are additional mechanisms by which genetic variants affect APA other than disruption of PAS motifs (S1C Fig).

As the PAS-interrupting allele (PIA) of PAS SNPs is expected to reduce cleavage efficiency at its p(A) site, we anticipated that the pA-QTL PIAs will be associated with lower pAUI (reflecting 3'UTR lengthening) (Fig 2A). However, unexpectedly, we also identified PAS

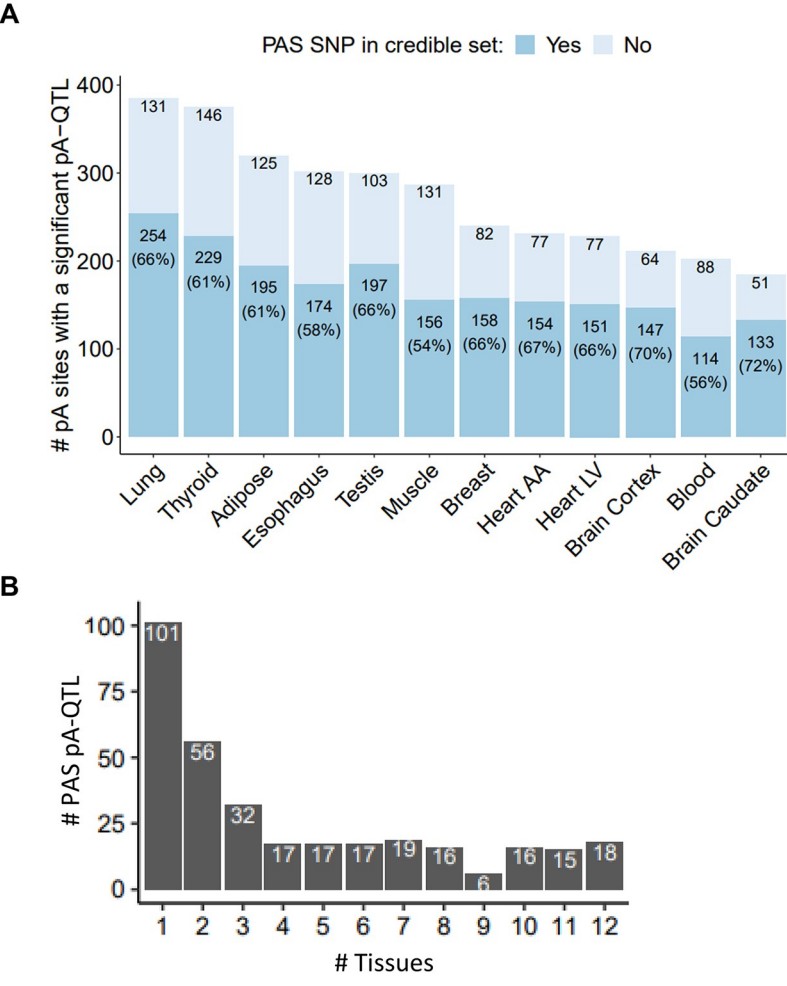

**Fig 3. PAS SNPs indicated by CAVIAR as putative causal pA-QTLs. A.** Bars showing the number of pA sites significantly associated with pA-QTLs per tissue, and the proportion of cases in which the corresponding PAS SNP is included in CAVIAR's credible set. **B.** Distribution of the number of different tissues in which each PAS SNP was indicated as pA-QTL variant. 49 PAS SNPs were indicated as causal pA-QTLs in at least 10 tissues.

pA-QTLs whose PIA was associated with increased pAUI (Fig 4A and 4B). Interestingly, in some of these cases, we observed that PIA's disruption of the cleavage at the corresponding p (A) site was associated with increased usage of an upstream (proximal) p(A) site (resulting in an increased pAUI and 3'UTR shortening). Yet, overall, the cases in which the PAI showed the expected 3'UTR lengthening effect were markedly more prevalent (84%-94%; Fig 4C). Furthermore, remarkably, for PAS pA-QTLs that were detected in multiple tissues, the effect of their PIA on 3'UTR length was highly consistent across all tissues (Fig 4D).

## Colocalization of PAS pA-QTL and eQTL signals

APA can impact gene expression via regulation of transcript stability. Therefore, to test for possible functional effects of PAS SNPs on gene expression (e.g., through regulation of transcript stability), we examined overlaps between PAS pA-QTL signals detected by our analysis and eQTL signals identified by GTEx (Fig 5A). Overall, of the 330 PAS pA-QTLs detected by our analysis, 133 overlapped an eQTL in at least one tissue. However, given the high

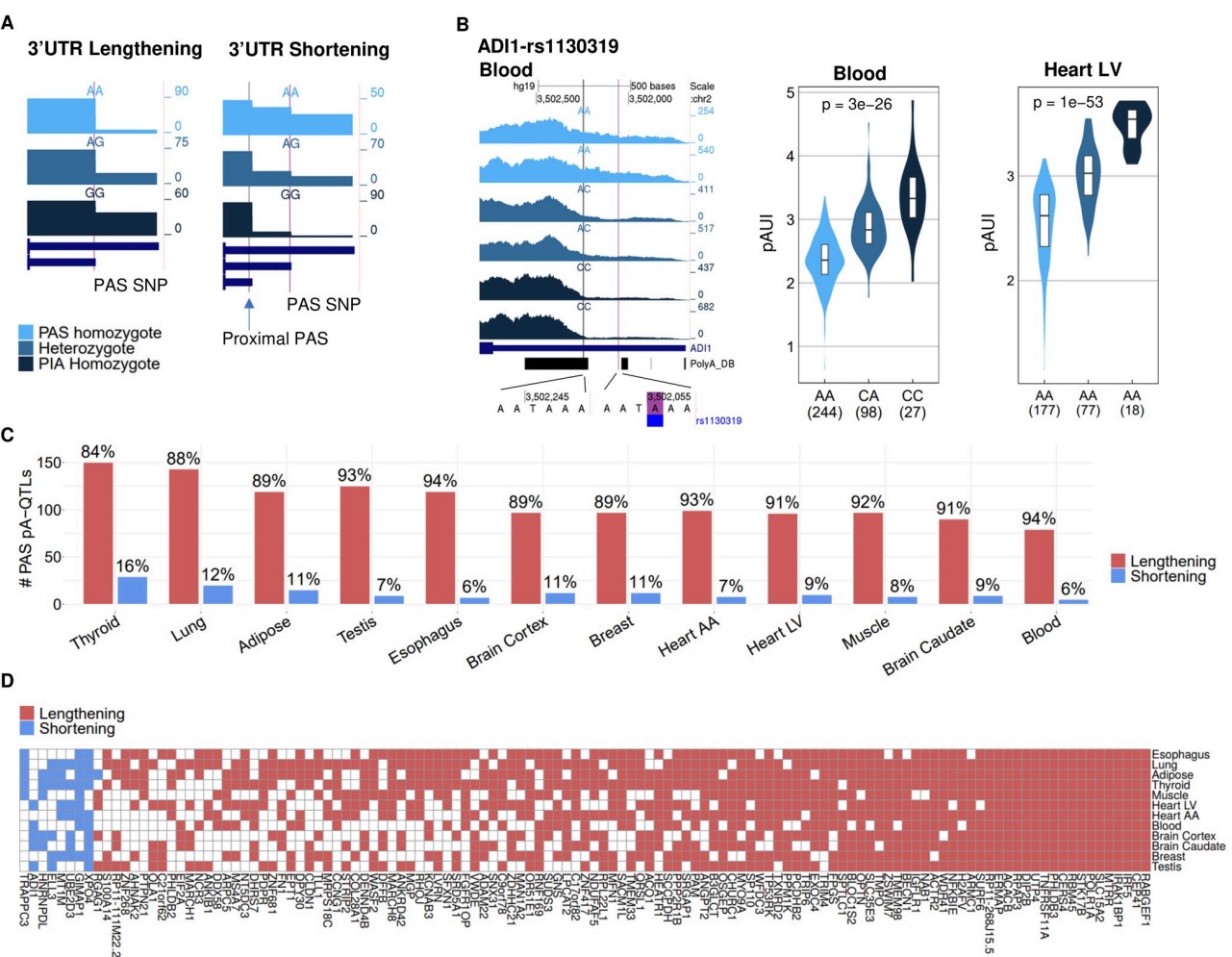

**Fig 4. The effect of PAS interrupting alleles (PIAs) on 3'UTR length. A**. Cartoons illustrating the anticipated 3'UTR lengthening effect of PIAs (left) and the unexpected 3'UTR shortening effect, due to elevated usage of an alternative proximal p(A) site (right). Note that in the lengthening case the PIA is associated with decreased pAUI levels whereas in the shortening case, the PIA is associated with elevated pAUI levels. **B**. An example of a PAS pA-QTL (rs1130319 in the 3'UTR of *ADI1*) whose PIA is associated with 3'UTR shortening (increased pAUI). Notably, this PAS SNP is detected as a pA-QTL in five different tissues, and in all these cases its PIA is consistently associated with 3'UTR shortening effect (shown in **D**) (nominal p-values obtained using FastQTL linear regression as described in the Methods). **C**. A bar chart of the effect of PAS pA-QTL's PIAs on 3'UTR length per tissue. As expected, in the vast majority of cases the PIA showed a lengthening effect. **D**. PIA effect on 3'UTR length. Shown are all PAS pA-QTLs detected in at least five tissues. Remarkably, in all these cases, the PIAs showed a consistent effect over all the tissues in which its PAS SNP was detected as a pA-QTL.

abundance of eQTLs in the human genome, a simple intersection between PAS pA-QTLs and eQTLs might result in many false hypotheses on the causal effect of PAS SNPs on gene expression [31]. Therefore, for loci in which a PAS pA-QTL intersected an eQTL, we used eCAVIAR [32] to seek further support for these two signals tagging the same causal variant (Methods; S2 Fig). Of the PAS pA-QTLs that overlapped an eQTL, 51 showed a high colocalization in at least one tissue (S3 Table).

Next, we examined a possible association between the effect of PIAs on 3'UTR length (lengthening or shortening) and gene expression (increased or decreased expression). As 3'UTR cis-regulatory elements mostly have destabilizing effects (e.g., ARE elements, micro-RNA binding sites), we expected that PIAs associated with 3'UTR lengthening will

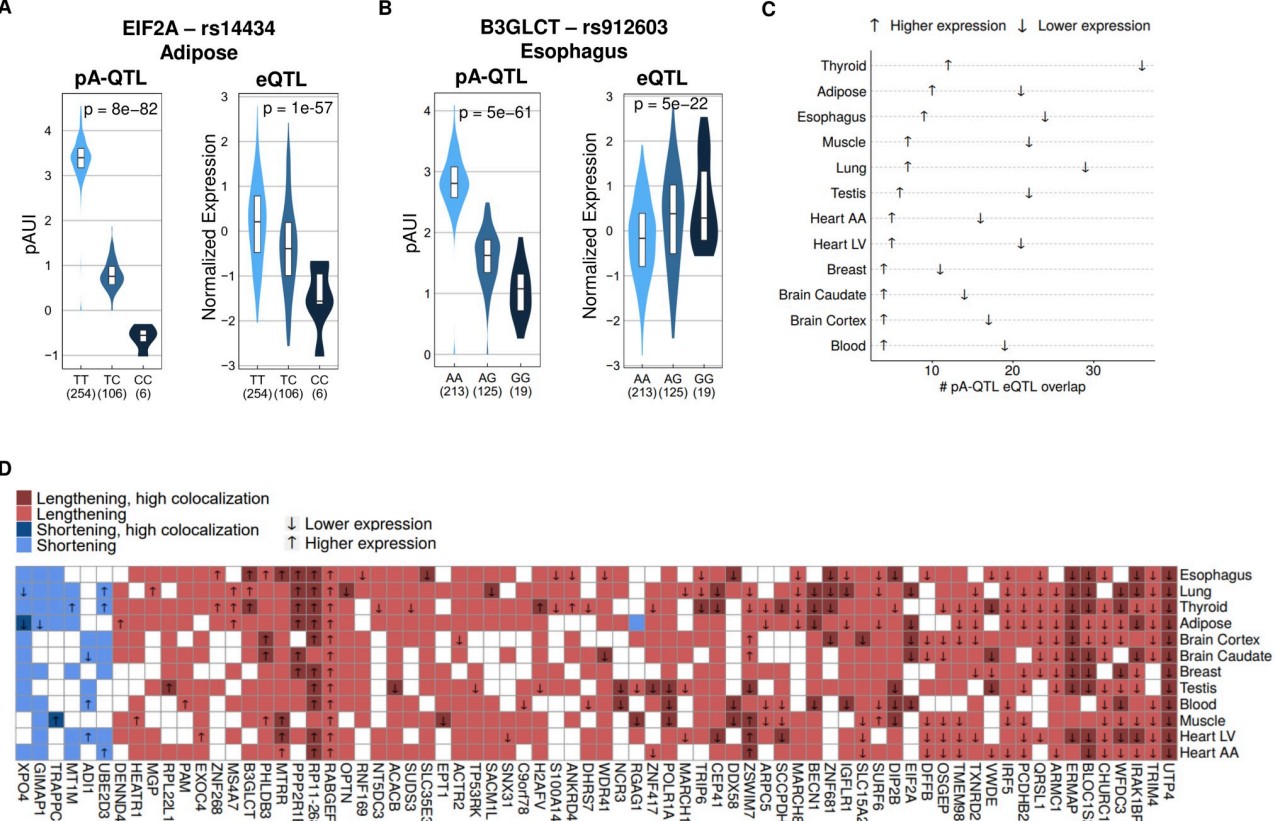

**Fig 5. Colocalization of pA-QTL and eQTL signals. A**. An example of a PAS SNP (rs14434 in the 3'UTR of *EIF2A*) that is both a pA-QTL and an eQTL (of this gene). The PIA of rs14434 (which is the C allele) is associated with lower pAUI at the corresponding p(A) site (and thus, 3'UTR lengthening) and lower expression level of *EIF2A*. Notably, rs14434 consistently showed this same effect in five different tissues (**Fig 5D**). **B.** An example of the uncommon case where a PIA (the G allele, in this case) was associated with decreased pAUI of the corresponding p(A) site (that is, 3'UTR lengthening) but with higher expression of the target gene. This PIA showed the same effect in three different tissues (**Fig 5D**) (pA-QTL nominal p-values calculated using FastQTL linear regression as described in the Methods, eQTL p-values were obtained from GTEx v7). **C**. A Cleveland dot plot of the PAS pA-QTLs overlapping an eQTL (for the same gene) whose PIA showed a 3'UTR lengthening effect. Arrow indicates the direction of the link between the PIA and gene expression. In all tissues, 3'UTR lengthening was significantly associated with decreased expression (one-tailed binomial tests, p-values < 0.05 in all tissues). **D.** Association between PIA effect on 3'UTR length (coded by color) and gene expression (shown by an arrow). Cases supported by the colocalization of the pA-QTL and eQTL signals (CLPP > 0.01) are shown in darker colors. (Shown in this heatmap are the PAS pA-QTLs with lengthening/shortening effect in at least seven tissues and that overlapped a GTEx eQTL in at least one tissue. Squares with no color indicate no overlap with eQTL).

consequently be mainly associated with decreased expression level (Fig 5A). Yet, we detected few uncommon cases in which 3'UTR lengthening effect of a PIA was associated with increased gene expression level (see Fig 5B for one example). Yet overall, in line with our expectations, in all tissues 3'UTR lengthening was significantly associated with decreased expression than with increased one (Fig 5C). Notably, both patterns of association with expression level were highly consistent across different tissues (Fig 5D). For example, the PIA of the PAS SNP in the 3'UTR of *UTP4* showed significant association with 3'UTR lengthening and decreased expression level in all twelve tissues, and the PIA of the PAS SNP in the 3'UTR of *PPP2R1B* showed a consistent association with 3'UTR lengthening and increased expression levels in six different tissues (Fig 5D). PIAs linked with 3'UTR shortening were not prevalent enough to allow robust statistical testing of their association with increased or decreased expression (Fig 5D).

## Colocalization of PAS pA-QTL and GWAS signals

Next, we sought possible links between the PAS pA-QTLs we identified and human pheno-types by intersecting these variants with GWAS risk SNPs (Methods). We analyzed GWAS summary statistics from 124 studies (S4 Table). In total, 104 PAS pA-QTLs overlapped at least one GWAS SNP (70 of which had GWAS lead SNP with p-value < 5e-8 and the rest had 5e-8 ≤ p-value < 1e-5), in 237 (158 when considering only GWAS lead SNP with p-value < 5e-8) PAS pA-QTL-GWAS SNP pairs. However, similar to eQTLs, given the high abundance of GWAS SNPs, pA-QTLs and GWAS SNPs can simply co-occur by chance [31]. Therefore, here too, we applied eCAVIAR to gain higher confidence for the potential causal effect of the PAS pA-QTLs on the complex traits. Overall, 78 (40) PAS pA-QTL and GWAS signals (from 37 (18) distinct PAS pA-QTLs) showed marked colocalization (S3 Fig and S3 Table). Nine (six) of these PAS pA-QTLs in nine (six) distinct genes were also colocalized with eQTL signals, sug-gesting novel links between APA and phenotype which are mediated through modulation of gene expression. These cases included the PAS pA-QTL in the 3'UTR of *BECN1*, a key regula-tor of autophagy. This pA-QTL was colocalized both with eQTL signals for *BECN1* and GWAS signals for type 2 diabetes (Fig 6A). Notably, autophagy abnormality has been recently associated with metabolic disorders, such as type 2 diabetes, and the BECN1 protein was shown to regulate insulin secretion. It was demonstrated that, in insulin-producing ß cells, excess autophagy degrades insulin-containing vesicles, resulting in decreased insulin contents and systemic glucose intolerance; whereas in insulin-responsive cells, activating autophagy decreases endoplasmic reticulum (ER) stress and improves insulin sensitivity [33]. Interest-ingly, the PIA of this PAS SNP is associated with 3'UTR lengthening, decreased *BECN1* expres-sion and decreased T2D risk (S3 Table). Other examples are the PAS pA-QTL in the 3'UTR of *PPP2R1B* that is colocalized with an eQTL for this gene and with GWAS signal of body fat dis-tribution (Fig 6B; interestingly, *PPP2R1B* shows outstanding high expression in fat pad tissues (Entrez Gene page)); and the PAS pA-QTL in the 3'UTR of *DIP2B* that is colocalized with an eQTL for this gene and with GWAS signal of red blood cell width (Fig 6C). Of note, our analy-sis did not capture the known association between the PAS SNP in the 3'UTR of IRF5 and risk for SLE [25], because this locus likely contains several causal variants that act through different mechanisms, some of which have a stronger effect on IRF5 expression and SLE risk than the PAS pA-QTL (S4 Fig). Accordingly, this PAS pA-QTL showed colocalization with the GWAS signal only after increasing eCAVIAR's setting for the number of putative causal variants in this locus to five (and with the eQTL signal after increasing the number of putative causal vari-ants to three) (S5 Table).

## pA-QTL analysis of 3'UTR SNPs

The above analyses were focused on SNPs that interrupt the canonical PAS signals, as those are expected to exert the strongest effect on APA. Yet, APA is regulated by additional 3'UTR cis-elements, including upstream U-rich and UGUA motifs, downstream U-rich and GU-rich elements and multiple RNA-binding sites [1]. Genetic variants that interfere with such auxil-iary elements are expected too to modulate cleavage efficiency at their respective p(A) sites, albeit with weaker impact. Therefore, last, we extended our analysis and searched for 3'UTR SNPs that are associated with APA modulation without disrupting PAS elements. The pAUI, which is based on pA site annotations, allows a high-resolution quantification of the usage of its PAS pA site. To analyze the effect of other variants on APA and consequently on 3'UTR length, we turned to using the distal p(A) site usage indexes provided by the APA Atlas [34] (Methods), as indexes for the overall (relative) 3'UTR length in each RNA-seq sample. This genome-wide 3'UTR pA-QTL analysis (covering 25,293 3'UTRs from 22,903 genes) identified

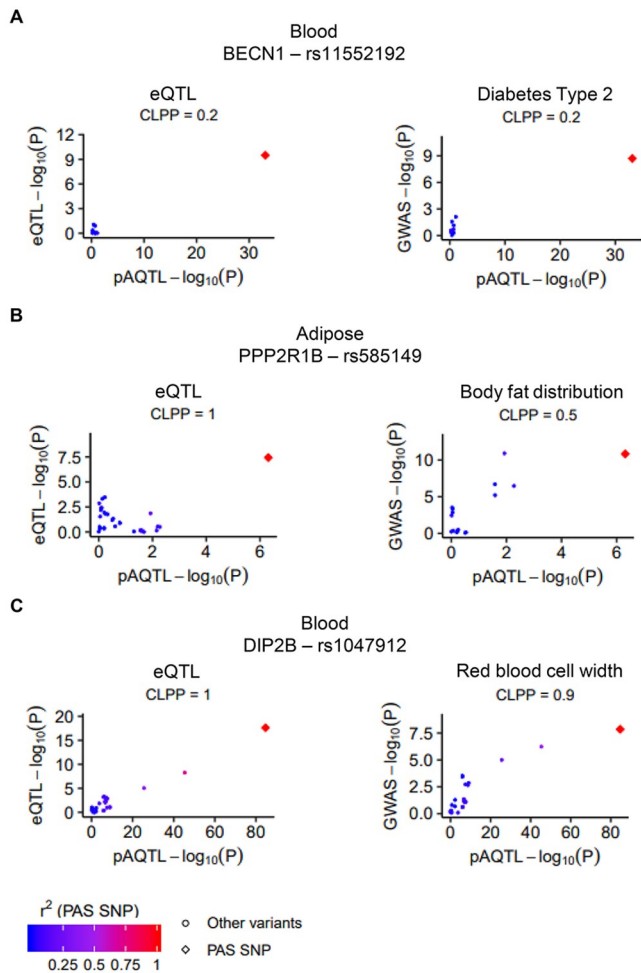

**Fig 6. Colocalization of pA-QTL, eQTL and GWAS signals.** Examples of PAS pA-QTLs that showed marked colocalization with both eQTL and GWAS signals in the 3'UTRs of the genes: **A**. *BECN1*. **B**. *PPP2R1B*. **C**. *DIP2B*. Dots are colored according to their LD ($r^2$) with the PAS SNP (calculated according to GTEx VCF files for eQTL plots and according to genome 1000 VCFs files for GWAS plots). Diamond shape signifies the PAS SNP. CLPP is colocalization posterior probability calculated using eCaviar (Methods).

27,994 pA-QTLs in 3,653 3'UTRs (3,501 genes) (and after excluding 3'UTRs containing PAS SNPs, 26,197 pA-QTLs in 3,577 3'UTRs of 3,429 genes). As anticipated, these pA-QTLs showed weaker effect sizes than the PAS pA-QTLs (Fig 7A). These additional pA-QTLs may modulate APA via multiple, direct and indirect, mechanisms. To further characterize one potential mechanism, we found 41 pA-QTLs (in 40 genes), supported by CAVIAR as likely tagging causal variants, that interfered with the GUGU motif, located within 20 nt downstream of their respective p(A) sites (Fig 7B; S6 Table). The effect sizes of these GUGU pA-QTLs were too markedly lower than those of the PAS pA-QTLs (S5 Fig). Unlike the sharp tendency for 3'UTR lengthening effect exerted by the PAS pA-QTLs (Fig 4C), as a set the GUGU-interfering pA-QTLs did not generally show a clear directional effect on 3'UTR length (Fig 7C). Yet, individually they showed a consistent effect across different tissues (Fig 7D). For example, the pA-QTL disrupting the GUGU motif in the 3'UTR of *TXN* was associated with 3'UTR lengthening in all twelve tissues (Fig 7D). Colocalization analysis of pA-QTL and eQTL signals

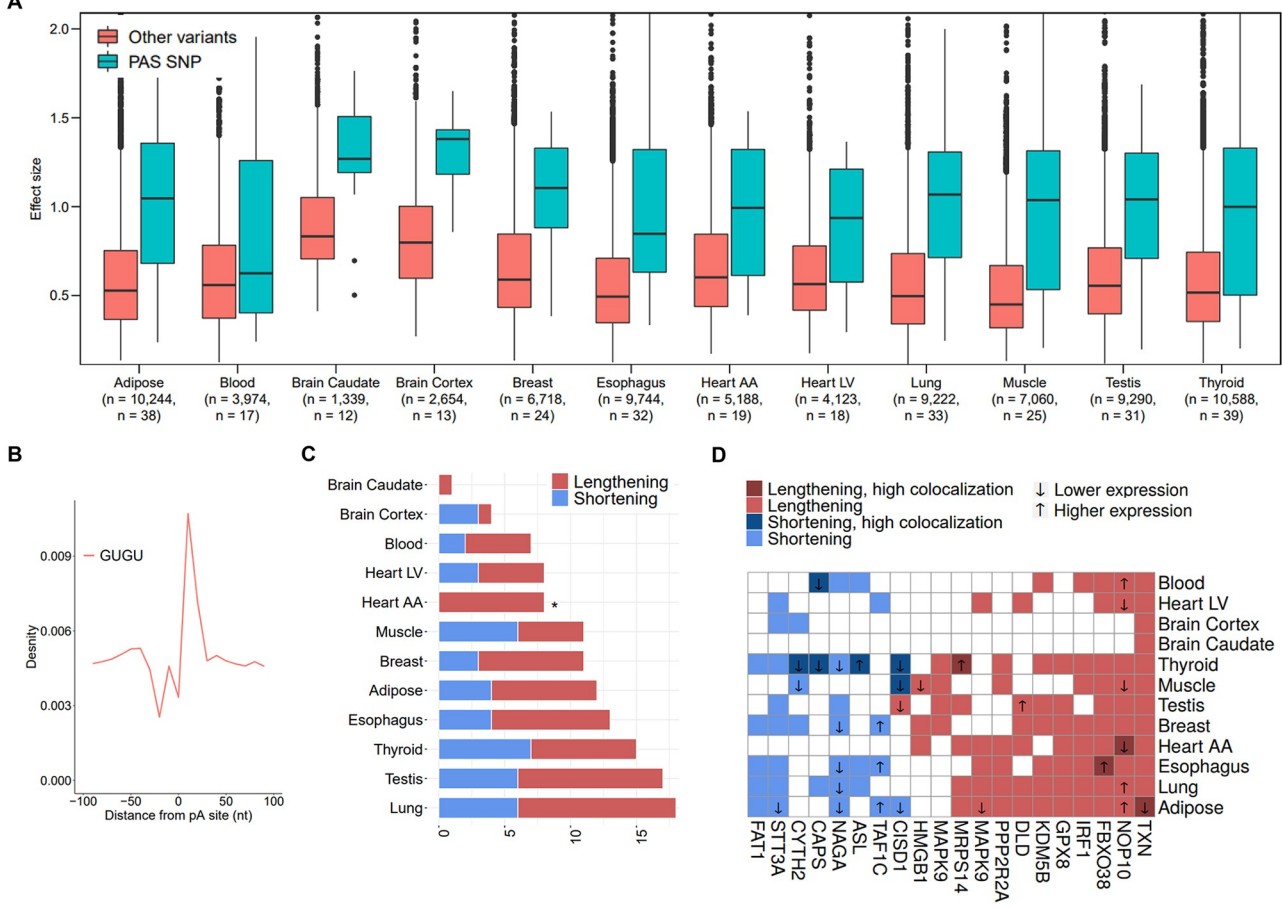

**Fig 7. Genome-wide pA-QTL analysis of 3'UTR SNPs. A.** Comparison between effect magnitudes (the absolute value of the slope calculated by FastQTL) of PAS pA-QTLs and other 3'UTR pA-QTLs (for both sets, we did not require here inclusion in CAVIAR's credible set). **B.** Location distribution of the GUGU motif with respect to 3'UTR p(A) sites (annotated in polyA DB). This motif shows a strong peak at ~20 nt downstream of the cleavage site. **C.** The effect of pA-QTLs interrupting a GUGU motif on 3'UTR length. (This analysis included the subset of these variants that were contained in CAVIAR's credible set; *p-value<0.05; calculated using a one-tailed binomial test). **D.** Association between pA-QTLs interrupting a GUGU motif, 3'UTR length and gene expression (eQTLs). Shown here are variants detected as pA-QTLs in at least three tissues. Colors are as in Fig 5D.

supported some of these GUGU-interfering SNPs as variants underlying both associations (Fig 7D; S6 Table).

## Discussion

Despite the emergence of APA as an important layer of gene regulation, potentially affecting the majority of human protein-coding genes, it remains largely unexplored, and its involvement in physiological and pathological processes is often overlooked. Here, we searched for SNPs that modulate APA. Focusing on SNPs likely to have the strongest effect, we identified 2,305 SNPs that interfere with the canonical PAS or its main variant. Seeking support for the regulatory impact of these SNPs on APA, we implemented pA-QTL tests using GTEx RNA-seq data from twelve tissues. Our analysis detected 330 PAS pA-QTLs with a significant effect in at least one tissue, which was supported by CAVIAR's fine-mapping as tagging causal variants. As expected, for the vast majority of these pA-QTLs, the PAS-interrupting allele caused

decreased cleavage at its p(A) site that resulted in 3'UTR lengthening (Fig 4C). Yet, interestingly, in ~10% of the cases, the effect of the PAS-interrupting allele was associated with 3'UTR shortening, due to elevated usage of a more proximal p(A) site (Fig 4A and 4B). This observation argues against a scanning mechanism as the mode of action of the polyadenylation machinery. The fact that reduced efficiency of a p(A) site augments cleavage and polyadenylation at an upstream p(A) site implies a thermodynamic competition between the alternative p(A) sites.

Our tests for the identification of PAS pA-QTLs were applied to each tissue separately. Yet, many of these variants showed a consistent effect across different tissues. Modern statistical methods strive to increase power by sharing information across conditions. Indeed, applying MASH, a recently developed method for joint multivariable analyses [35], to the set of PAS SNPs markedly increased the number of PAS SNPs detected as pA-QTLs across multiple tissues (S6A and S6C Fig). Nevertheless, attention should be paid not only to statistical significance but also to effect sizes. Notably, the PAS SNPs which, in the MASH analysis, had a significant effect on APA in all twelve tissues, showed high heterogeneity in the magnitude of their effect over the tissues (S6B Fig), similar to results recently observed for eQTLs [35].

APA potentially affects post-transcriptional gene regulation in multiple ways, including modulation of mRNA stability, translation efficiency, nuclear export, and cellular localization. However, two recent studies failed to detect a large effect of APA on transcript stability or translation efficiency [36, 37]. We sought an indication of the functional impact of the PAS SNPs on gene expression. To this goal, we intersected the PAS pA-QTLs identified by our analysis with GTEx eQTLs and found that 133 of them, in at least one tissue, were also detected as eQTLs of the same gene. Noting that given the high abundance of eQTLs in the human genome, such overlaps are likely to occur by chance, we further used eCAVIAR to examine colocalization of the pA-QTL and eQTL signals. This analysis indicated 51 PAS SNPs as underlying causal variants for the association with both APA and gene expression. As cis-regulatory elements embedded within 3'UTRs mainly carry destabilizing roles (e.g., miRNA binding sites, AU-rich elements (AREs)) [38], we expected that 3'UTR lengthening effects of the PAS-interrupting alleles will be mostly linked with reduced gene expression. We indeed observed such link in all twelve tissue (Fig 5C). These results indicate the regulatory impact of APA on gene expression, where shorter isoforms generated by APA are generally more stable than longer ones. Nevertheless, the impact of 3'UTR APA on transcript stability is more complicated than mere inclusion/exclusion of regulatory elements in/from the 3'UTR, since the efficiency of mRNA targeting by such elements can be also affected by their location, as was demonstrated for miRNA target sites: sites located at the start or end of the 3'UTRs are more efficient than those located in the middle [39]. Thus, APA can modulate the activity of miRNA target sites by changing their location relative to the transcript's 3' end [40].

Over the last decade, GWAS studies discovered thousands of SNPs associated with common diseases and traits (The GWAS catalog already reports >70k tag SNPs [41]). Yet, the mechanism of action of most of the genetic variants identified by GWAS is currently unknown. Functional interpretation is hindered by the fact that the vast majority (>90%) of these SNPs map to noncoding regions of the genome [42]. While disruption of enhancer elements regulating gene transcription emerges as the main mode of action of risk SNPs [42], marked fractions of traits' heritability are not accounted for by SNPs that map to transcriptional regulatory elements (e.g., putative enhancers and promoters) [43]. This indicates that other modes of action mediate the impact of noncoding genetic variants on human traits. Modulation of APA can be an important additional mechanism, and in this study, we identified 104 PAS pA-QTLs that overlapped GWAS SNPs (in 237 pA-QTL-GWAS trait pairs). However, similar to eQTLs, GWAS SNPs are too highly abundant in the human genome, and

thus such overlaps may occur just by chance. To gain higher confidence in the potential causal effect of the PAS pA-QTLs on the complex human traits, we applied eCAVIAR, and detected marked colocalization of the pA-QTL and GWAS signals for 78 pairs in 37 distinct 3'UTR loci. Our analysis mainly focused on PAS SNPs as they are expected to have the strongest impact on APA (and consequently, on gene expression and human traits). Nevertheless, our extended, analysis of 3'UTR pA-QTLs suggests that thousands of additional variants effect (although with weaker magnitude) gene expression and human phenotypes by APA modulation. Very recently, Yang et al. [44] carried out a similar pA-QTL analysis on 32 human cancer types using RNA-seq data from The Cancer Genome Atlas (TCGA) and their study too detected thousands of cis pA-QTLs.

Given the critical roles potentially played by APA in gene regulation and our limited understanding of how it is affected by genetic variants, our methodology and findings contribute to the initial elucidation of associations between PAS SNPs, gene expression and human phenotypes.

## Methods

### Identification of PAS SNPs

To identify PAS SNPs, we started with all 106,571 p(A) sites located in 3'UTRs from poly(A) DB (release 3.2) [27], and all 37,611,962 annotated SNPs from GTEX v7 [28]. We considered the 53,428 SNPs located within 40-nt upstream of an annotated 3'UTR p(A) site. Examining the sequences spanning 5-nt upstream and 5-nt downstream of these SNP, we identified 2,260 instances in which one of the alleles (reference or alternative) constitute a canonic PAS sequence (AATAAA) or its main variant (ATTAAA), while the other allele interrupts this signal (we call this allele the PAS-interrupting allele—*PIA*). We also detected 45 instances in which one allele constitutes the canonic PAS, while the other allele constitutes the main variant.

### GTEx data

Aligned GTEx paired-end RNA-seq were obtained from dbGaP (release phs000424.v7.p2). We used SAMtools [45] and SRA tools to convert cram and SRA files, respectively, to BAM format. We used RNA-seq data from 12 tissues that have more than 130 RNA-seq samples with corresponding genotype data (S2 Table). SNP genotypes were obtained from GTEx v7 VCF file, which is based on whole-genome sequencing (we used VCF processed files from which low-quality sites and samples are filtered out by GTEx).

### pA-QTL analysis of PAS SNPs

We associated each PAS SNP with its p(A) site (located within 40-nt downstream of it; if a PAS SNP had more than one annotated p(A) site within downstream 40-nt, the nearest p(A) site was taken). We split the 3'UTRs containing a PAS SNP into two segments: the common 3'UTR (*cUTR*; the segment upstream of the p(A) site) and the alternative 3'UTR (*aUTR*; the 3'UTR segment downstream of the p(A) site). 3'UTR coordinates were downloaded from the UCSC genome browser based on GENCODE hg19 release v31 [46]. Overlapping 3'UTRs of the same gene were merged using bedtools [47]. When the p(A) site of a PAS SNP was located at the edge of the 3'UTR, the entire 3'UTR was considered as the cUTR and the downstream 1,000 nt were considered as the aUTR.

We defined the pA site Usage Index (*pAUI*) to quantify the usage of each p(A) site associated with a PAS SNP in each RNA-seq sample:

$$pAUI = \log_2\left(\frac{cUTR\ counts + 1}{aUTR\ counts + 1}\right),$$

where *cUTR counts* is the fragment counts in the cUTR, and *aUTR counts* is the fragment counts in the aUTR. A pseudo count of 1 was added to avoid zeroes.

We used featureCounts [48] with a SAF file containing the coordinates of the 3'UTR segments to count the number of reads mapping to aUTRs and cUTRs in each sample. Sequenced fragments that overlapped a 3'UTR were considered in the following way: if at least one of the fragment's reads intersected the aUTR region, the fragment was assigned to the aUTR; otherwise, the fragment was assigned to the cUTR segment. Only reads that were uniquely mapped (mapping quality of 255), aligned in proper pairs, and not marked as PCR duplicates, were counted.

We next used the pAUI levels for a pA-QTL analysis. We carried out this analysis for each tissue separately. To ensure sufficient statistical power while lowering the burden of multiple testing, for each tissue, we included in our analysis only 3'UTRs with a median RPKM $\geq 1$ across samples and corresponded to genes that were included in GTEx v7 analysis (namely genes with $> 0.1$ TPM and $\geq 6$ reads in 20% of the samples). We performed the pA-QTL analysis using FastQTL [29]. We added sequencing platform (Illumina HiSeq 2000 or HiSeq X), sex, the top three genotype principal components and PEER factors (obtained from the GTEx data portal) as covariates. For each tissue and each 3'UTR containing a PAS SNP, we examined all variants located within 10 kbp of the corresponding pA site whose minor allele frequencies $\geq$0.01 and with the minor allele observed in at least 10 samples. To account for testing multiple (correlated) variants within each pA site window, we followed the procedure applied by GTEx v7 [28]. Briefly, we ran FastQTL with the permutation mode to obtain empirical p-values extrapolated from a Beta distribution (FastQTL setting–permute 1000 10000). This mode reports the p-value of the most significant variant in a locus (pA site, in our analysis). These p-values were further corrected for multiple testing (due to testing pA sites over multiple 3'UTRs) using Storey's q-value method [49]. pA sites with q-value $< 0.05$ were considered significantly associated with a pA-QTL.

For each significant pA locus, we applied CAVIAR [30] to define the 95% credible set, which contains the causal variant with high confidence. The inputs to CAVIAR for each set were variants' t-statistics (calculated from the slope and slope standard error from FastQTL) and their pairwise LD (*r*) calculated using PLINK 1.9 (—r square). (PLINK's binary file was created from GTEx VCF using plink—make-bed and specifying—keep allele-order). Since we suspect that in each tested pA locus the PAS SNP is the causal variant, CAVIAR was run with one causal variant mode (-c 1) and otherwise default parameters. We considered a PAS SNP as a likely causal pA-QTL (referred to as a PAS pA-QTL) if (a) it was included in the credible set of its pA locus and (b) its nominal p-value was below its pA-locus threshold calculated following GTEx's procedure for obtaining a gene-level nominal p-value threshold based on its beta distribution model [28].

All tests were performed in R-3.5.1 and plots were made using ggplot2 R package.

## Colocalization of pA-QTL and eQTL signals

For each tissue, we first intersected its GTEx v7 list of significant eQTLs (significant variant-gene pairs obtained from GTEx .signif_variant_gene_pairs.txt files) with its list of PAS pA-QTLs (requiring that these eQTLs are linked to the genes in which they are located). We

then used eCAVIAR [32] to test these cases for colocalization of the eQTL and pA-QTL signals. (The strength of colocalization between two genomic signals is measured by eCAVAIR's colocalization posterior probability (CLPP)). For each pair of a PAS pA-QTL and its overlapping eQTL, eCAVIAR analysis included variants located within +/- 10 kbps with respect to the corresponding pA site (filtered as described above for the pA-QTL analysis). eCAVIAR input per p(A) locus were the variants' t-statistics obtained by the pA-QTL and eQTL tests and pairwise LD scores calculated as described above for CAVIAR analysis. eCAVIAR was run using the 1 causal variant mode (-c 1), and otherwise, default parameters. We followed eCAVIAR authors' practice and considered CLPP > 0.01 as an indication for colocalization.

## Colocalization of pA-QTL and GWAS signals

We used summary statistics from 124 GWAS studies (S4 Table) to find overlaps between PAS pA-QTLs and significant GWAS SNPs (p-value for a given trait < 1e-5, we also distinguish in S3 Table cases for which the lead GWAS SNP in the locus had p-value < 5e-8). Similar to the analysis of colocalization between pA-QTLs and eQTLs, here too we used eCAVIAR to seek further support for the pA-QTLs and GWAS SNPs tagging the same causal variants. For each such overlap, the input to eCAVIAR was the pA-QTL's t-statistics and GWAS' z-scores of variants included in both datasets and that are located within +/- 10 kbp of the respective pA site, and LDs for GTEx and GWAS calculated from GTEx VCF file and 1000 genome VCF files respectively.

As eCAVIAR requires the direction of effect and direction of LD, we aligned the GWAS z-scores such that the reference allele (non-effect allele) is the same as the GTEx reference allele. Furthermore, to align the pA-QTL eQTL LDs, we used BCF tools [50] to match the reference alleles of the genome 1000 VCFs to those of GTEx (After using BCF tools to split multiallelic sites to biallelic). We then used PLINK specifying -keep allele-order when creating PLINK binary files (this step is required as by default, PLINK 1.9 uses the major allele as the reference). eCAVIAR was run using the 1 causal variant mode (-c 1), and otherwise, default parameters.

## Genome-wide pA-QTL analysis of 3'UTR SNPs

In this analysis, we used the distal p(A) site usage index provided by the APA-Atlas [34], calculated per transcript in each GTEx sample (transcripts with less than 30 reads in their last exon were removed by APA-atlas). We included in this analysis all variants located within 3'UTRs or their 2.5 kbp flanking regions (GENCODE hg19 release v31) whose minor allele frequencies ≥0.01 and with the minor allele observed in at least 10 samples. To identify pA-QTLs, each 3'UTR interval was analyzed by FastQTL, using the same procedure and the same covariates described above for the analysis of PAS SNPs. Variants (a) located in 3'UTR loci with q-value<0.05 and (b) obtained nominal p-value below the interval-specific threshold were considered significant 3'UTR pA-QTLs. CAVIAR and eCAVIAR analyses were applied to the significant variants that interrupt GUGU element as described above for the PAS pA-QTLs.

## MASH

We used MASH (multivariate adaptive shrinkage) [35] as implemented by mashr. Analyzing 3'UTRs with PAS SNPs, the inputs were the matrix of effect estimates and the matrix of corresponding standard errors (slope and slope standard error calculated by FastQTL) of the set of 26,140 variants that passed the variants' filtering criteria (see above) in all tissue. We fitted the MASH model using both data-driven and canonical covariance as recommended by the authors. To obtain the data-driven covariance, we used mashr built-in functions (*get_significant_results*, *cov_pca*). In this analysis we considered a PAS SNP as a pA-QTL PAS if (a) its local false sign rate (*lsfr*) was below 0.05, and (b) it was included in CAVIAR's credible set.

CAVIAR was run as described above, with FastQTL t-statistics replaced by the ratio of the posterior mean and posterior standard error obtained from MASH.

Raw data for the figures and supplemental tables of this study are available at https://github.com/ElkonLab/pA_QTLs.

## Supporting information

**S1 Fig. CAVIAR fine-mapping analysis. A**. The credible sets that were defined by CAVIAR for the pA loci presented in Fig 2B–2D. **B**. Examples for PAS SNPs with very high posterior probability pointing them as the causal pA-QTL variants with very high confidence. **C**. Examples of PAS SNPs that obtained a significant p-value in the pA-QTL tests, but were not included in the credible sets of their pA loci, indicating that they are not the causal ones. Varaints included in the credible set are colored in green. Otherwise, the legend is as in Fig 6.
(TIF)

**S2 Fig. eCAVIAR colocalization analysis of pA-QTL and eQTL signals. A**. Examples of loci showing strong colocalization of the pA-QTL and eQTL signals (very high colocalization posterior probability (CLPP)), indicating the PAS SNP as the causal variant for both effects. **B**. Examples of weak colocalization of the pA-QTL and eQTL signals (low CLPP), suggesting that the eQTL tags a causal variant that is distinct from the PAS SNP. **C**. Strong colocalization of the pA-QTL and eQTL signals in the loci shown in Fig 5A and 5B. Legend is as in Fig 6.
(TIF)

**S3 Fig. eCAVIAR colocalization analysis of pA-QTL and GWAS signals.** Examples of PAS pA-QTLs and GWAS SNPs showing high colocalization (CLPP>0.01), indicating a common underlying causal mechanism. Legend is as in Fig 6.
(TIF)

**S4 Fig. Relationship between pA-QTL, eQTL and SLE GWAS signals in the 3'UTR locus of *IRF5*.** Legend is as in Fig 6, See S6 Table.
(TIF)

**S5 Fig. Comparison between effect magnitudes of PAS pA-QTLs, GUGU pA-QTL and other 3'UTR pA-QTLs.** Axis and legend as Fig 7A.
(TIF)

**S6 Fig. MASH information sharing across conditions increases statistical power for the detection of pA-QTL effects. A**. Distribution of the number of different tissues in which each PAS SNP pA-QTL was detected by the tissue-by-tissue tests (left) and by MASH analysis utilizing information sharing across tissues (the analysis shown here includes variants that were tested (that is, passed the filtering criteria) in all tissues). The proportion of PAS SNPs that were detected as pA-QTLs in at least nine tissues was up from 20% in the tissue-by-tissue tests to 45% in MASH analysis. **B.** PAS SNPs with a significant effect in all twelve tissues (39 SNPs) show high variation in their effect sizes over the tissues. For each PAS SNP detected by MASH as a pA-QTL in all tissues, we calculate the ratio between its maximal and minimal effect size (MASH posterior mean). The violin plot shows the distribution of these ratios. A third of these pA QTLs show more than a 2-fold difference in their effect size. **C**. An example of a PAS pA-QTL that reached statistical significance in only two tissues (Thyroid and Blood) by the tissue-by-tissue FastQTL tests, and turned significant in ten tissues by MASH analysis based on information sharing across tissues. (The indicated p-values are those obtained by the tissue-by-tissue FastQTL tests, asterisks next to the tissue name indicate pA-QTL according to MASH analysis).
(TIF)

**S1 Table. 3'UTR SNPs interrupting a canonical PAS (AATAAA) or its main variant (ATTAAA).**
(XLSX)

**S2 Table. GTEx tissues analyzed in our study.**
(XLSX)

**S3 Table. PAS pA-QTLs and their association with eQTLs and GWAS SNPs.**
(XLSX)

**S4 Table. GWAS summary statistics datasets analyzed in our study.**
(XLSX)

**S5 Table. SLE IRF5 eCaviar analysis results.**
(XLSX)

**S6 Table. GUGU-interfering variants and their association with eQTLs.**
(XLSX)

## Author Contributions

**Conceptualization:** Eldad David Shulman, Ran Elkon.

**Data curation:** Eldad David Shulman.

**Formal analysis:** Eldad David Shulman.

**Methodology:** Eldad David Shulman, Ran Elkon.

**Writing – original draft:** Eldad David Shulman, Ran Elkon.

**Writing – review & editing:** Eldad David Shulman, Ran Elkon.

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
