## [Decision Letter · Decision Letter 0]

10 Jan 2020

Dear Dr Elkon,

Thank you very much for submitting your Research Article entitled 'Systematic identification of functional SNPs interrupting 3' UTR polyadenylation signals' to PLOS Genetics.

The manuscript was fully evaluated at the editorial level and by two independent peer reviewers. As you will see, the reviewers (and the editors) believe that this work addresses a timely question which appears to be of high interest to the broad genetics community and has to great potential to significantly advance the field. However, one of the reviewers has raised major points that need to be addressed before the manuscript can be considered for publication.  Based on the reviews and our editorial evaluation, we will not be able to accept this version of the manuscript, but we would be willing to review again a much-revised version. We cannot, of course, promise publication at that time.

If you decide to revise the manuscript for further consideration at PLOS Genetics, please aim to resubmit within the next 60 days, unless it will take extra time to address the concerns of the reviewers, in which case we would appreciate an expected resubmission date by email to plosgenetics@plos.org.

[LINK]

We are sorry that we cannot be more positive about your manuscript at this stage. Please do not hesitate to contact us if you have any concerns or questions.

Yours sincerely,

Andreas Gruber

Guest Editor

PLOS Genetics

Gregory Barsh

Editor-in-Chief

PLOS Genetics

Reviewer's Responses to Questions

**Comments to the Authors:**

Reviewer #1: The manuscript “Systematic identification of functional SNPs interrupting 3’ UTR polyadenylation signals” by Shulman and Elkon examines genome-wide, the effect of SNPs that interrupt the polyadenylation signal (PAS) in the 3’ UTR on differential poly(A) site usage using RNA-seq data from 12 tissues in GTEx through QTL analysis (pA-QTL) and their relationship with complex disease or trait associations. The authors focus on variants that fall in canonical PAS within 40 nucleotides of the poly(A) tail, which are expected to have the strongest effects on polyA. They find 139 significant pA-QTLs, half of which overlap with expression QTLs (eQTLs), and 53 of which are in linkage disequilibrium (LD) with complex disease or trait associations, proposing a new (under investigated) molecular mechanism that may underlie genetic associations with complex diseases/traits. One finding the authors find unexpected is that 12% of variants in the polyA motif that lead to 3’ UTR lengthening are associated with increased expression of the gene, while 88% of cases are associated with decrease in expression. Transcript lengthening has primarily been thought to have a destabilizing effect.

This work presents afirst systematic, genome-wide examination of variants in PAS association with differential polyadenylation levels in various tissues, providing new insights on the effect of genetic variation on transcript expression and human complex traits. While I think such a resource would be very valuable to the genetic community, I have several technical comments/concerns and suggestions on the analyses which I think need to be addressed for the results to be sound and more broadly applicable.

Major comments:

1. The authors performed QTL (association) analysis between SNPs found in polyadenylation signals (PAS) within 40 nucleotides of annotated 3’ UTR polyadenylation (p(A)) sites and p(A) site usage based on reads that map to the common 3’ UTR segment (upstream of the pA site) relative to the reads that map to the alternative 3’ UTR segment (downstream of the pA site). While this is a first obvious region to inspect for genetic effects on polyadenylation, as disruption of the canonical PAS has been shown to have strong effect on pA usage, there may be other cis effects on polyadenylation (e.g., binding of RNA-binding protein to the DNA). I think it would be informative to test for genetic associations with pA usage considering also variants further away from the 3’ UTR, e.g., within +/-1Mb of the pA site (this would identify other types of mechanisms that may affect pA usage).

2. Also, in computing pA site usage, did the authors apply a lower bound cutoff of number of reads that map to the 3‘ UTR or lower bound density of reads per Kb?

3. If a PAS SNP had more than one annotated p(A) site within downstream 40 nucleotides, the nearest pA site was taken. Why not test all adjacent pA sites and correct for this in the permutation analysis when assessing significance of associations per gene?

4. There is no mention of adding hidden expression covariates (e.g., PEER factors, Stegle et al., 2012) to the linear regression model of the pA-QTL analysis. It is very important to add expression covariates, as for example the first expression PEER factors in GTEx are highly correlated with ischemic time that affect RNA degradation, which is a factor that can affect the pA usage, or cellular composition heterogeneity. The PEER factors are provided on the GTEx portal (https://www.gtexportal.org/home/) for all tissues.

5. For the pA-QTL analysis the authors use a minor allele frequency cutoff of >0.1. At the sample size of over 130 samples there is power to detect associations at a minor allele frequency of 0.05.

6. Can the authors expand on how they computed q-values from the QTL association p-values? Is this using permutation analysis or fitting a beta distribution? Was the FDR applied at the gene level, e.g. taking the best signal per gene? A description of how to properly account for differences in number of variants tested per gene and local LD can be found in GTEx v6 paper (PMID: 29022597) or on the GTEx portal.

7. To inspect the connection between the effect of the pA-QTLs and genetic associations with complex human diseases and traits, the authors test for overlap/co-occurrence of signals. However, co-occurrence of QTL and GWAS signals may just be by chance, given the clustering of GWAS signals around genes and their high abundance (PMID: 31043754). Bayesian-based colocalization methods have been developed to test whether two co-occurring association signals are tagging the same causal variant or haplotype, such as eCAVIAR (PMID: 27866706) and ENLOC (PMID: 28278150) which consider local allelic heterogeneity. I would recommend applying such methods to gain higher confidence in the potential causal effect of pA-QTLs on complex human traits as for example done in GTEx release v8 preprint (https://www.biorxiv.org/content/10.1101/787903v1).

Also, it would be informative to report the direction of effect of pA-QTLs on disease risk or trait variation for those pA-QTLs that colocalize with GWAS associations? Does lengthening or shortening of the transcript tend to be associated with increased risk of disease?

8. To test whether the 139 pA-QTLs were also significant eQTLs, the authors tested whether the pA-QTL variants were also significant eQTLs. However, similarly to GWAS associations, pA-QTL and eQTL signals may be overlapping but tagging different causal variants. To further support that the pA-QTL and are also eQTLs, I would also recommend applying colocalization analysis between co-occurring pA-QTLs and eQTLs,

9. To better estimate sharing of pA-QTLs across tissues, there are multi-tissue methods that account for effect size and standard error across tissues, to increase power in each tissue. The multivariate adaptive shrinkage method, MASH (PMID: 30478440) is one such method that estimates tissue sharing and specificity of genetic associations.

Minor comments:

1. 139 PAS SNPs were found to be significant pA-QTLs at FDR<5%. For how many genes? Was only one variant association found to be significant per gene?

Reviewer #2: This is a nicely carried out work on SNP impacts on APA. This is also a timely piece of work, because several studies are reporting similar findings. One key finding of this work is the relationship between 3’UTR size control and gene expression. I have only a few minor comments:

1. Have the authors tried a different window for SNP analysis than 40 nt? It would be helpful to readers if they can show SNPs in more upstream or downstream regions.

2. A sizable fraction of APA-affecting SNPs do not involve canonical signals. The authors need to show some of the samples and discuss them.

**Have all data underlying the figures and results presented in the manuscript been provided?**

Reviewer #1: None

Reviewer #2: Yes

PLOS authors have the option to publish the peer review history of their article (what does this mean?). If published, this will include your full peer review and any attached files.

Reviewer #1: No

Reviewer #2: No

---

## [Decision Letter · Decision Letter 1]

29 Apr 2020

Dear Dr Elkon,

Thank you very much for submitting your Research Article entitled 'Systematic identification of functional SNPs interrupting 3' UTR polyadenylation signals' to PLOS Genetics. Your manuscript was fully evaluated at the editorial level and by independent peer reviewers. The first reviewer had some additional minor points that need to be addressed, whereas the second reviewer happy that you have fully addressed all his concerns.

We, the editors are convinced that your study is of very high interest to the community and substantially advances the 3' end processing field. That's why we would be happy to publish your manuscript, given that the additional minor issues raised by Reviewer 1 can be addressed.

We therefore ask you to modify the manuscript according to the review recommendations before we can consider your manuscript for acceptance. Your revisions should address the specific points made by each reviewer.

[LINK]

Yours sincerely,

Andreas Gruber

Guest Editor

PLOS Genetics

Gregory Barsh

Editor-in-Chief

PLOS Genetics

Reviewer's Responses to Questions

**Comments to the Authors:**

Reviewer #1: The authors of the manuscript “Systematic identification of functional SNPs interrupting 3’ UTR polyadenylation signals” have done a lot of work to the revised version and have carefully addressed all of my comments/concerns. I only have a few remaining minor comments:

1. To test whether the significant pA-QTL colocalize with GWAS signals, the authors tested 124 complex diseases and traits with available summary statistics. They tested for significant colocalization between PAS pA-QTL with GWAS p-values below 1E-05 (Methods, page 13). Given the multiple hypothesis burden of testing millions of common variants in GWAS, the cutoff P<5E-08 has been typically used as the genome-wide significant cutoff. I would recommend to distinguish between GWAS loci whose lead variant passes genome-wide significance and those that are subthreshold associations (5E-08<p<1e-05), analysis.="" colocalization="" consider="" example="" for="" genome-wide="" gwas="" in="" just="" or="" signals="" significant="" the="" to="">

2. On the top of page 6, the authors point out that they did not capture the known association between the PAS SNP in the 3'UTR of IRF5 and risk for SLE when setting the eCAVIAR parameter for number of putative causal variants to 1, “because this locus likely contains several causal variants that act through different mechanisms, some of which have a stronger effect on IRF5 expression and SLE risk than the PAS pA-QTL (S4 Fig).” This indeed seems to be the case. The authors go on to test this by setting the parameter to multiple potential causal variants in the SLE GWAS and eQTL loci, and report that then they do find significant colocalization with 3-5 putative causal variants. Can the authors add these results to a supplementary table? What are the posterior probabilities of the lead GWAS and pA-QTL variants?

3. On the bottom of page 5, in the sentence: “Overall, 512 pA sites were significantly associated (FDR=5%) with a pA-QTL in at least one tissue (see examples in Fig 2B-D; S3 Table).” it should be written FDR<=5%, not (FDR=5%).

4. On the bottom of page 6, the authors report the results of testing for colocalization of co-occurring PAS pA-QTLs and eQTLs, also in Table S3: “Of the PAS pA-QTLs that overlapped an eQTL, 51 showed a high colocalization in at least one tissue (S3 Table).” I think it would be helpful to generate a table that summarizes just these 51 significantly colocalizing results, to be able to easily view these results over the thousands of tests performed across all tissues.

5. On the top of page 7 and in Figure 5C, the authors note that 3'UTR lengthening in all tissues is significantly associated with decreased expression of the target gene. Is this for all overlapping pA-QTL and eQTL or just for the 51 significantly colocalizing loci? If the latter, it would be clearer to write ‘Colocalizing pA-QTL / eQTL loci’ in the x-axis of Fig. 5C.

6. In figures 5D and 7D, arrows that indicate the direction of the link between the PIA and gene expression were added only to a subset of the genes by tissue squares. Were they added only to significantly colocalizing pA-QTL/eQTL loci? If so, I would recommend clarifying this in the figure legends.

7. On the top of page 6, I would add the words “per tissue” at the end of the following sentence: “Defining per pA site with a significant association a 95% credible set of variants, this analysis implicated the PAS SNP as a causal variant for 55%-70% of the observed pA-QTLs (Fig3A).”

8. In the Methods section on the bottom of page 12 and on pages 13 and 14, the authors use the term ‘t-score’ to represent the regression slope divided by its standard error. I think it is more correct to call this value ‘t-statistic’.

9. There is a typo on the bottom of page 13: ‘GETx’ should be ‘GTEx’.

10. There is a typo on the top of page 14: -keep alle-order should be ‘-keep allele-order’.

11. There is a typo in the word ‘slop’ (should be ‘slope’) in the MASH paragraph in the Methods section on the bottom of page 14.

Reviewer #2: The authors have addressed all my concerns.</p<1e-05),>

**Have all data underlying the figures and results presented in the manuscript been provided?**

Reviewer #1: Yes

Reviewer #2: Yes

PLOS authors have the option to publish the peer review history of their article (what does this mean?). If published, this will include your full peer review and any attached files.

Reviewer #1: No

Reviewer #2: No

---

## [Decision Letter · Decision Letter 2]

1 Jul 2020

Dear Dr Elkon,

We are pleased to inform you that also reviewer 1 has now accepted your revised work. Thus we are very happy to accept your manuscript entitled "Systematic identification of functional SNPs interrupting 3' UTR polyadenylation signals" for publication in PLOS Genetics. Congratulations!

Yours sincerely,

Andreas Gruber

Guest Editor

PLOS Genetics

Gregory Barsh

Editor-in-Chief

PLOS Genetics

Comments from the reviewers (if applicable):

Reviewer's Responses to Questions

**Comments to the Authors:**

Reviewer #1: The authors have addressed all of my comments satisfactorily. I have no additional comments. The manuscript is ready for publication.

**Have all data underlying the figures and results presented in the manuscript been provided?**

Reviewer #1: Yes

PLOS authors have the option to publish the peer review history of their article (what does this mean?). If published, this will include your full peer review and any attached files.

Reviewer #1: No

**Data Deposition**

http://datadryad.org/submit?journalID=pgenetics&manu=PGENETICS-D-19-01596R2

**Press Queries**

---

## [Editor Report · Acceptance letter]

10 Aug 2020

PGENETICS-D-19-01596R2 

Systematic identification of functional SNPs interrupting 3'UTR polyadenylation signals 

Dear Dr Elkon, 

We are pleased to inform you that your manuscript entitled "Systematic identification of functional SNPs interrupting 3'UTR polyadenylation signals" has been formally accepted for publication in PLOS Genetics! Your manuscript is now with our production department and you will be notified of the publication date in due course.

With kind regards,

Kaitlin Butler

PLOS Genetics

On behalf of:
